# Hybrid 1D + 2D Modelling for the Assessment of the Heat Transfer in the EU DEMO Water-Cooled Lithium-Lead Manifolds

**Antonio Froio [1,*], Andrea Bertinetti [1,2], Alessandro Del Nevo [3] and Laura Savoldi [4]**

[1]    NEMO Group, Dipartimento Energia "Galileo Ferraris", Politecnico di Torino, 10129 Torino, Italy; andrea.bertinetti@polito.it

[2]    Gemmate Technologies SRL, 10090 Buttigliera Alta (TO), Italy; andrea.bertinetti@gemmate-technologies.com

[3]    ENEA FSN-ING-SIS, CR Brasimone, 40032 Camugnano (BO), Italy; alessandro.delnevo@enea.it

[4]    MAHTEP Group, Dipartimento Energia "Galileo Ferraris", Politecnico di Torino, 10129 Torino, Italy; laura.savoldi@polito.it

*    Correspondence: antonio.froio@polito.it; Tel.: +39-011-090-4494

**Abstract:** The European demonstration fusion power reactor (EU DEMO) tokamak will be the first European fusion device to produce electricity and to include a breeding blanket (BB). In the framework of the design of the EU DEMO BB, the analysis of the heat transfer between the inlet and outlet manifold of the coolant is needed, to assess the actual cooling capability of the water entering the cooling channels, as well as the actual coolant outlet temperature from the machine. The complex, fully three-dimensional conjugate heat transfer problem is reduced here with a novel approach to a simpler one, decoupling the longitudinal and transverse scales for the heat transport by developing correlations for a conductive heat-transfer problem. While in the longitudinal direction a standard 1D model for the heat transport by fluid advection is adopted, a set of 2D finite elements analyses are run in the transverse direction, in order to lump the 2D heat conduction effects in suitable correlations. Such correlations are implemented in a 1D finite volume model with the 1D GEneral Tokamak THErmal-hydraulic Model (GETTHEM) code (Politecnico di Torino, Torino, Italy); the proposed approach thus reduces the 3D problem to a 1D one, allowing a parametric evaluation of the heat transfer in the entire blanket with a reduced computational cost. The deviation from nominal inlet and outlet temperature values, for the case of the Water-Cooled Lithium-Lead BB concept, is found to be always below 1.4 K and, in some cases, even to be beneficial. Consequently, the heat transfer among the manifolds at different temperatures can be safely (and conservatively) neglected.

**Keywords:** nuclear fusion; EU DEMO; breeding blanket; water-cooled lithium-lead (WCLL); thermal-hydraulics; system-level modelling

## 1. Introduction

The European demonstration fusion power reactor (EU DEMO) reactor (see Figure 1 [1]) aims at becoming the first European tokamak fusion reactor to produce net electrical energy and provide it to the grid [2]. Therefore, as a main difference with respect to ITER [3], it will be part of the first generation of tokamaks to include a fully functional breeding blanket (BB), having shielding, breeding and power extraction functions [4]. Within the EUROfusion Consortium, responsible for the development of the EU DEMO, different BB concepts are being investigated [4], one of which is the Water-Cooled Lithium-Lead (WCLL) [5], employing liquid PbLi (eutectic) as breeder and neutron multiplier material, and water as coolant.

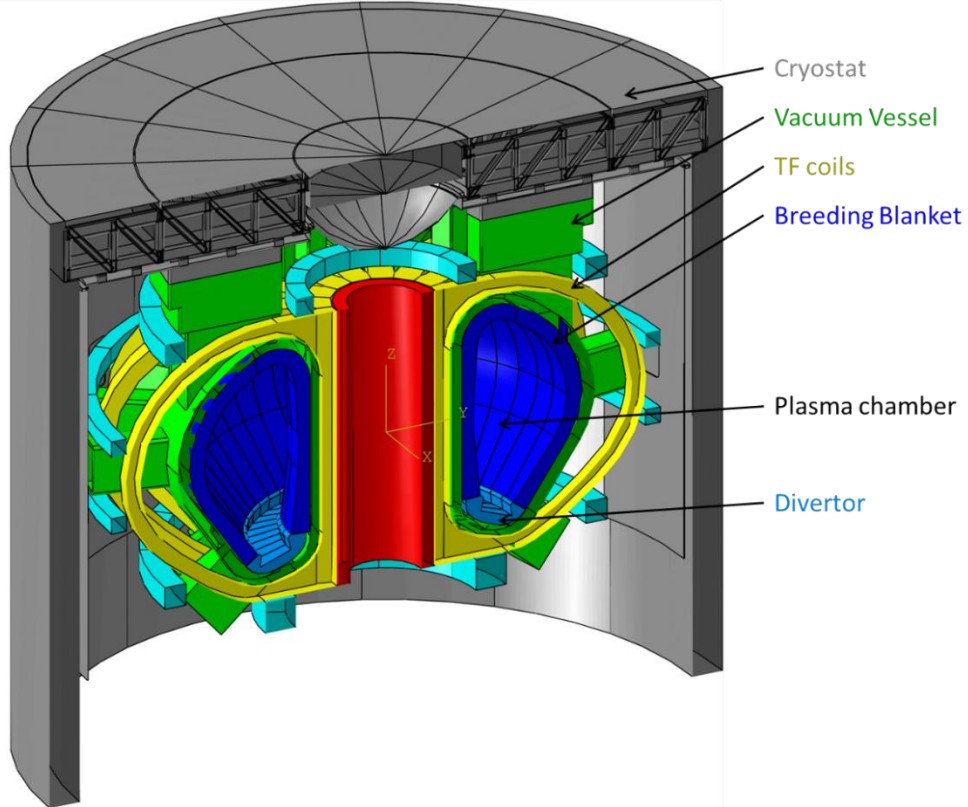

**Figure 1.** The European demonstration fusion power reactor (EU DEMO) tokamak (Reproduced from [1], EUROfusion Consortium: 2020).

The WCLL (see Figure 2 [6]) is a promising concept of BB for the EU DEMO, whose design is led by the Italian National Agency for New Technologies, Energy and Sustainable Economic Development (ENEA). The design team includes several Italian universities (Sapienza University of Rome, the CREATE consortium at the Federico II University of Naples, University of Pisa, University of Palermo and Politecnico di Torino), plus collaboration with international research institutes. To breed the tritium needed for the fusion reactions, the WCLL foresees the use of molten Pb-16Li, slowly circulating inside the different BB segments. As coolant, a forced flow of pressurized water (15.5 MPa, 295–328 °C) is used, allowing the exploitation of know-how from fission pressurized water reactors [7].

In the 2019 design of the WCLL back supporting structure (BSS), the interface with the balance-of-plant (BoP) is at the upper port, i.e., the coolant enters the machine from the top part of Figure 2a. The distribution and collection of the liquid breeder and coolant inside the BB segment is undertaken by means of manifolds located in the back part of the segment (i.e., the furthest from the plasma), also known as the BSS, see Figure 2b where the BSS is highlighted in green. To guarantee adequate distribution of the coolant at the different poloidal locations, the cross section of the coolant manifolds varies along the BSS.

The coolant enters the machine from two different circuits, one for the first wall (FW) and one for the breeding zone (BZ). Consequently, in the BSS there are different channels (see Figure 2d):

- Inlet manifold for the FW cooling channels (labelled as "FW inlet" in Figure 2d)
- Outlet manifold for the FW cooling channels (labelled as "FW outlet" in Figure 2d)
- Inlet manifold for the BZ cooling tubes (labelled as "BZ inlet" in Figure 2d)
- Outlet manifold for the BZ cooling tubes (labelled as "BZ outlet" in Figure 2d)
- Coaxial PbLi inlet and outlet manifolds (labelled "PbLi" in Figure 2d)

The coolant enters the corresponding inlet manifold and flows downwards in the poloidal direction, from where it is delivered to the different FW cooling channels or double-walled tubes (DWT), used to cool the BZ (i.e., the region where most of the tritium is generated). From the channels or tubes, the coolant flows back in the corresponding outlet manifold in the BSS, where it flows upwards in the poloidal direction, until it leaves the machine. The coolant flow path is reported in Figure 2d. The BB is segmented in the poloidal direction in "elementary units," separated by the toroido-radial stiffening plates, all sharing the same structure as reported in Figure 2c. Each elementary unit includes 10 FW channels and 20 BZ DWTs, see Figure 2c. The elementary unit is toroidally symmetric with respect to its midplane.

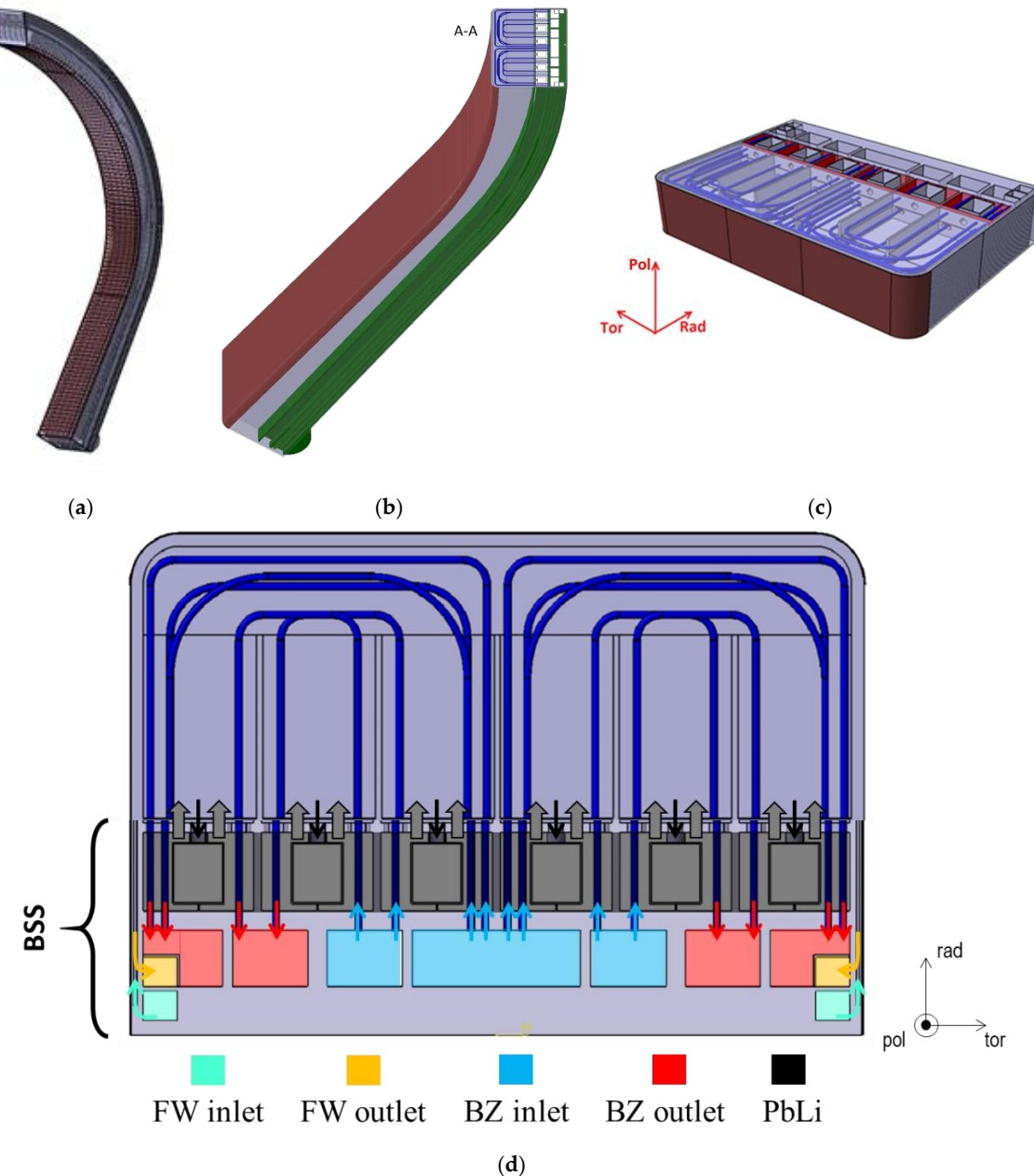

**Figure 2.** The 2019 WCLL BB: (**a**) isometric view of a central outboard breeding blanket (BB) segment; (**b**) lateral view, highlighting the back-supporting structure (BSS) in green, with a section at the equatorial plane, showing the details of the internals; (**c**) isometric view of a single elementary unit; (**d**) toroidal-radial view of a single elementary unit at the equatorial outboard midplane, showing the details of the coolant flow from/to the manifolds. (Adapted from [6], EUROfusion Consortium: 2020).

As a consequence of this flow path, in the BSS region, fluids at different temperatures are found: liquid PbLi at temperature of about 330 °C, "cold" (inlet) water at 295 °C and "hot" (outlet) water at 328 °C. As one of the functions of the BB is to extract the power produced in the reactor, it is important that the outlet temperature of the BB coolant is as close as possible to the design value. A reduction of such temperature would imply a reduction of the overall plant efficiency. Also, the coolant inlet temperature to the different FW channels and DWTs, should be as close as possible to the design value: a temperature increase there would translate into a higher temperature of the structural materials. Considering the geometry of the manifold region, however, some heat transfer between the "cold" and "hot" water is expected to take place. In fact, inlet and outlet manifolds are separated by thin metallic walls, made of the structural material, i.e., the reduced-activation ferritic-martensitic EUROFER97 steel, see Figure 2d.

The design of the BB cannot exploit experience from previous generations of tokamak reactors, and extrapolation from current or future experiments is limited. The expertise that will be acquired from the ITER test blanket modules (TBM [4]) will not be sufficient due to both the very different size of the TBM, with respect to a full BB, and to the timescale envisaged for the ITER deuterium-tritium experimental campaigns (currently foreseen after 2036 [8]). Dedicated experiments are foreseen, but their applicability is limited to smaller scales and to single effects (i.e., integral experiments cannot be pursued). As a consequence, the DEMO BB design must be thoroughly supported by computational tools, that will be qualified with properly scaled mock-ups and experimental campaigns. Such computational tools include detailed, three-dimensional (3D) models for the neutronics, thermal-mechanics and computational fluid-dynamics (CFD). These codes are complemented by fast-running system-level codes, which can provide answers at the global level in a relatively short time. For the thermal-hydraulic modelling, such codes include e.g. RELAP (Reactor Excursion and Leak Analysis Program) [9] and MELCOR (MELt CORe) [10], which are system-level tools widely applied in the nuclear (fission) field, or the GEneral Tokamak THErmal-hydraulic Model (GETTHEM), which is a tool developed specifically for tokamak fusion reactors at Politecnico di Torino [11]. The GETTHEM code has been applied in the past to both normal operating conditions for the Helium-Cooled Pebble Bed [12], the WCLL BZ [13] and FW [14] cooling systems, as well as for accidental transient analyses, including both gaseous helium [15] and two-phase water [16] as working fluids.

System-level codes use a lumped-parameter approach, where complex (usually 3D) objects are simplified to 1D, or lumped into a 0D object. This is often obtained from analytical formulas, whenever available, or deducing correlations or constitutive relations from a set of more complex analyses. The use of correlations to reduce the dimensionality of a problem and achieve a faster solution is a well-established technique in thermal-hydraulics, in particular for hydraulic systems where there is a single, main direction of fluid motion (such as flows in pipes, channels or ducts). The most common application of correlations is to determine the "friction factor" coefficient for the calculation of distributed pressure drops in transitional and turbulent flows, for which the most notable example is the Colebrook (or Colebrook-White) correlation [17]. Such correlations are generally obtained by regression of experimental results, and are thus applicable only within the range of validity of the used data. Often, for specific applications (e.g., complex flow patterns), correlations existing in literature are not suitable, or have been proven experimentally not to provide acceptable results, so dedicated experimental campaigns may be set up with the aim of deriving new correlations. This is the case of, e.g., [18], where a correlation for the friction factor in the central channel of superconducting cables for fusion has been derived.

Similarly, another classical field of application of correlations is to lump the convective heat transfer into a lower-dimensionality model. As for the case of the friction factor, several correlations are available in literature, for different fluids, flow regimes and patterns, boundary conditions and configurations. Again, for those applications where existing correlations are inapplicable, dedicated experimental campaigns have been performed to derive new correlations, see e.g., [19] where heat transfer correlations are derived from experimental data on solar thermal receivers. Other multidimensional physical

phenomena are often reduced to correlations to allow implementation of simple models, e.g., in [20] correlations have been experimentally derived for direct contact condensation of steam jets in subcooled water (e.g., for suppression pools).

In case the experiments are too expensive or not available, due for instance to issues related to the materials to be handled (e.g., toxicity) or size constraints, numerical experiments based on computational models have also been used. The main advantages of exploiting computational models with respect to experiments are the reduced costs, the possibility to model the real-size object, and the easier and faster development. However, the computational models must be subject to a rigorous solution verification and validation [21] against some relevant experimental dataset, in order to confirm the modelling choices. Moreover, as the validation cannot be performed on the same set of data for which the correlation is derived, there is still a residual risk associated with extrapolation. Examples of computationally-derived correlations in literature include for instance the derivation of the friction factors in meander-flow heat exchangers in the "current leads" to feed the fusion magnets both in laminar [22] and turbulent [23] regimes. Three-dimensional CFD models have been widely used in different fields; some examples are the derivation of correlations for convective heat transfer of a helical coil heat exchanger [24], of building surfaces as a function of wind velocity [25], or in two-phase flow conditions in direct contact steam condensation in feedwater heaters [26]. In the solar energy field, a 3D CFD model at the microscale is used in [27] to characterize the smallest-scale component of a solar volumetric receiver, which is lumped using correlations for convective heat transfer in a system-level model [28]. In all cases above, both experimental and numerical, the focus is either on the friction factor or on <u>convective</u> heat transfer correlations, whereas <u>conductive</u> heat transfer is never lumped in correlations (to the best of the authors' knowledge).

In the present work, a hybrid numerical method is developed and applied to the evaluation of the heat transfer between the cold and hot manifolds of the EU DEMO BSS. A 2D finite element (FE), steady-state, pure thermal model of the BSS manifold cross section (in its 2019 design version) is first used to analyze the heat conduction effects in the septa dividing the cold and hot sides of the WCLL BSS manifold. These effects are lumped into thermal resistances, for which a set of correlations is derived. The correlations developed from the 2D model <u>across</u> the manifold are then implemented in a (transient) 1D finite volume (FV) model of the manifolds, where the direction considered is <u>along</u> the fluid direction. This 1D model is developed using the GETTHEM code. The 1D model is then applied to steady-state system-level thermal-hydraulic analyses of the WCLL, to quantify the heat transfer among the different manifolds in the BSS region. Indeed, the heat transfer among the different manifolds shall be minimized to avoid the reduction of the cooling capabilities (due to higher coolant temperature at the inlet of the cooling channels) and of the overall plant efficiency (due to lower coolant temperature at the inlet of the steam generator). To this end, the deviation from the nominal value of the mentioned temperature values is evaluated, with a parametric study on the BSS geometry. The main novelty of this work, with respect to the literature reported above, consists in the application of a higher-dimensionality code to lump a <u>conductive</u> heat transfer rather than a <u>convective</u> heat transfer. Indeed, conductive heat-transfer problems are generally solved directly as they are not particularly demanding in terms of computational cost, due to the nature of the equation to be solved. In conjugate heat-transfer problems (i.e., where both conduction and convection heat transfer take place), like the one at hand here, most of the time is spent by the computer solving the fluid-dynamic problem rather than the conduction problem. Nevertheless, as the fluid-dynamic problem is 1D in this case, the conduction problem (being fully 3D) would represent the largest contribution to the overall computational cost of the problem. The application of the approach proposed here enables the use of a 1D, system-level fluid-dynamic problem, further reducing the computational cost and enabling parametric studies, which are of utter importance during the design phase.

The paper is structured as follows: in Section 2 a brief description of the WCLL BSS is reported, followed by the description of the 2D FE and 1D GETTHEM FV models; in Section 3, the main results of the analysis are presented; finally, in Section 4 the results are discussed and some conclusions are drawn.

## 2. Materials and Methods

A possible approach to evaluate the negative effects of the by-pass heat transfer from the outlet (hot) manifold to the inlet (cold) one would be a fully 3D thermal-hydraulic (conjugate heat transfer) analysis of the entire BB segment. However, an analysis of that kind would require a strong computational effort, considering the different spatial scales (e.g., thickness of the walls in the range of millimetres, height of the segment in the range of ~10 m). To reduce the computational cost, in the present work the by-pass heat transfer is evaluated combining a 2D thermal model of the BSS to a 1D thermal-hydraulic model of the manifolds, which includes both fluid and solid. The adoption of a 2D model rather than a 3D model (including the poloidal direction) is justified considering that:

1.  The temperature gradient in the poloidal direction in the BSS solid regions will be negligible;
2.  The temperature in the other internals (i.e., outside the BSS) of the BB will be almost identical at all poloidal locations, as effect of the cooling (which is designed to keep an almost uniform temperature, to reduce thermal stresses) [6].

Consequently, as the temperature gradient is the driver of conductive heat transfer, in the poloidal direction such heat transfer will be negligible.

The 2D model, implemented using the FE FreeFem++ software (v4.3, F. Hecht, Paris, France) [29], solves the steady-state heat diffusion Equation (1) [30] with boundary conditions (BCs) as in Equations (2)–(5) [30]:

$$\begin{cases} \nabla \cdot (k\nabla T) = -q''' & in \; \Omega & (1) \\ T = T_0 & on \; \Gamma_D & (2) \\ -k\nabla T = q'' & on \; \Gamma_N & (3) \\ -k\nabla T = \left(T - T_g\right) & on \; \Gamma_{R,C} & (4) \\ -k\nabla T = f_v \in \sigma\left(T^4 - T_{VV}^4\right) & on \; \Gamma_{R,r} & (5) \end{cases}$$

where $k$ is the thermal conductivity, $T$ is the temperature, $q'''$ is the volumetric power generated in the domain $\Omega$, $T_0$ is the temperature imposed on the $\Gamma_D$ boundary (Dirichlet BCs), $\Gamma_N$ is the boundary where Neumann BCs are applied, $q''$ is the surface heat flux applied on $\Gamma_N$, $\Gamma_{R,c}$ is the boundary where convective Robin BCs are applied, $h$ is the convective heat transfer coefficient (HTC), $T_g$ is the fluid bulk temperature, $\Gamma_{R,r}$ is the boundary where radiative Robin BCs are applied, $f_v$ is the view factor, $\varepsilon$ is the surface emissivity, $\sigma$ is the Stefan–Boltzmann constant and $T_{VV}$ is the temperature of the surface facing the $\Gamma_{R,r}$ boundary. The FE method does not solve the problem in the strong formulation reported in Equation (1), but adopts the weak formulation [31], which naturally includes Neumann and Robin BCs, as reported in Equation (6):

$$\int_\Omega k\nabla T \cdot \nabla w d\omega + \int_{\Gamma_N} q'' w d\gamma + \int_{\Gamma_{R,c}} h\left(T - T_g\right) w d\gamma + \int_{\Gamma_{R,r}} f_v \epsilon \sigma\left(T^4 - T_{VV}^4\right) w d\gamma = \\ = \int_\Omega q''' w d\omega \quad \forall w \in H^1_{\Gamma_D}(\Omega),$$

$$(6)$$

where $w$ is the so-called "test function" and $H^1_{\Gamma_D}(\Omega)$ is a Sobolev subspace, i.e., it is the space of continuous, piecewise affine functions in $\Omega$, locally equal to $T_0$ on $\Gamma_D$.

The domain and BCs used in this work are depicted in Figure 3; half of the BSS region is modelled for symmetry reasons, and the 2D cut is taken at the equatorial outboard midplane. The PbLi manifolds are treated as convective BCs, as well as the water manifolds; the value of the HTC for the PbLi is computed according to the correlation for magnetohydrodynamic (MHD) flows reported in [32], see Equation (7):

$$Nu = \tfrac{2}{3}Nu_s + 0.015Pe^{0.8}, \tag{7}$$

where $Nu = hL/k$ is the Nusselt dimensionless number, $L$ is a characteristic length (half the channel size for MHD flows), $Nu_s = 4.8$ is the slug Nusselt number (i.e., the Nusselt number determined analytically assuming a flat velocity profile on the cross section), $Pe = RePr = \rho c_p L v/k$ is the Péclet dimensionless

number, $Re = \rho v L/\mu$ is the Reynolds dimensionless number, $Pr = c_{\text{p}}\mu/k$ is the Prandtl dimensionless number, $\rho$ is the fluid density, $c_{\text{p}}$ is the fluid specific heat at constant pressure, $v$ is the fluid average velocity and $\mu$ is the fluid dynamic viscosity. For the water HTC, the Gnielinski correlation [33] is used, see Equation (8):

$$Nu = \frac{f/8(Re - 1000)Pr}{1 + 12.7(f/8)^{1/2}(Pr^{2/3} - 1)}, \tag{8}$$

where $f$ is the Darcy friction factor ($L$ is here taken as the hydraulic diameter of the duct). The PbLi and water properties are evaluated as a function of the temperature and of pressure and temperature, respectively; for PbLi, the properties are taken from [34] and evaluated at the fixed temperature of 598 K (used also as bulk temperature), whereas for water the Industrial Formulation '97 (IF97) of the International Association for the Properties of Water and Steam (IAPWS) is used [35], assuming a fixed pressure of 15.5 MPa. For both water and PbLi, as the channel cross section varies along the poloidal direction, the characteristic length for the evaluation of the dimensionless numbers changes accordingly; as the 2D model is taken at the equatorial midplane (see Figure 2b–d), the corresponding values are assumed.

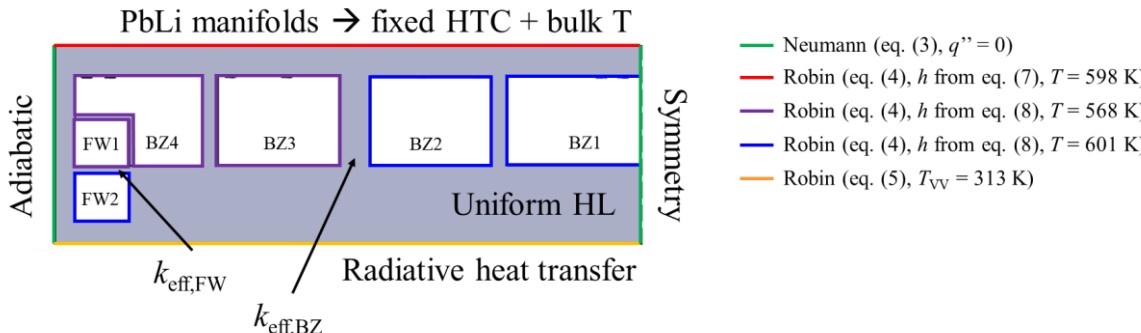

**Figure 3.** Computational domain and boundary conditions of the 2D finite element (FE) thermal model, highlighting with different colors the different boundary conditions; the nomenclature used in the 2D FE model is also reported. (HL: heat load).

For the wall facing the vacuum vessel (VV), a radiative heat transfer BC is assumed, following the Stefan–Boltzmann law reported in Equation (5). The view factor is conservatively set to 1, as well as the emissivity; the VV temperature $T_{\text{VV}}$ is fixed to 40 °C. To have an idea of the effect of the hypotheses on $f_{\text{v}}$ and $\varepsilon$, the model is run also setting the radiative heat transfer to zero, thus having an adiabatic BC. The introduction of the radiative heat transfer poses a nonlinearity in the problem, which is solved using fixed-point iterations up to a tolerance of $10^{-8}$. The thermal conductivity of the EUROFER97 is assumed constant and equal to 33.2 W/(m·K); this assumption is reasonable as the EUROFER97 thermal conductivity varies by about 0.5% in the temperature range of interest. Note that no heat flux coming from the plasma is assumed, as its contribution to the loading in the domain of interest is "filtered" by the presence of the FW, BZ and PbLi manifolds; the only BC on the plasma side is then the convective BC due to heat transfer to the PbLi manifolds.

As the final 1D GETTHEM model will lose information about the 2D effects of heat diffusion in the BSS, a set of correlations is derived for the thermal resistance in between two manifolds. These correlations are derived as a function of the wall thickness, which is then varied parametrically in the 2D model. Note that a parametric analysis of this kind does not require a strong computational effort, as it is limited to a 2D domain.

The thermal resistance $R$ between two manifolds is evaluated according to the definition in Equation (9) [30]:

$$R \triangleq \frac{\Delta T}{Q} = \frac{1}{k}\frac{t}{S}, \tag{9}$$

where $Q$ is the heat flowing through the resistance, $\Delta T$ is the temperature difference across the resistance, $t$ is the thickness of the material slab constituting the thermal resistance (see Figure 4), and $S$ is its surface area in the direction perpendicular to the heat flow. Note that this definition is rigorously valid under the assumption of a steady state (which is the case of this work), no internal heat generation and no radiative effects. As $R$ is computed on a 2D geometry (and thus the $S$ and $Q$ values should become infinite), rather than the actual value of the resistance, Equation (9) is multiplied by $S$, and an "effective thermal conductivity" $k_{\text{eff}}$ is computed, according to Equation (10):

$$k_{\text{eff}} = \frac{\phi}{\Delta \overline{T}} t, \tag{10}$$

where $\phi = Q/S$ is the heat flux flowing through the wall and $\Delta \overline{T}$ is the difference between the average temperature on the two surfaces of the wall, see Figure 4. Computing the heat flux and temperature difference on the solid surfaces, rather than considering the (known) fluid bulk temperature, makes the computed value of $k_{\text{eff}}$ independent on the convective HTC, so it is valid for any value of the fluid velocity in the manifold. Recall that the fluid velocity varies continuously along the manifold axis, due to continuous removal (addition) of water from the inlet manifolds (to the outlet manifolds), as well as to the variation of the manifold dimensions. To verify this assumption, the $k_{\text{eff}}$ values have been computed varying the water HTC in the Robin BCs in the range ±20% of the value obtained with Equation (8), see Section 3.1 below.

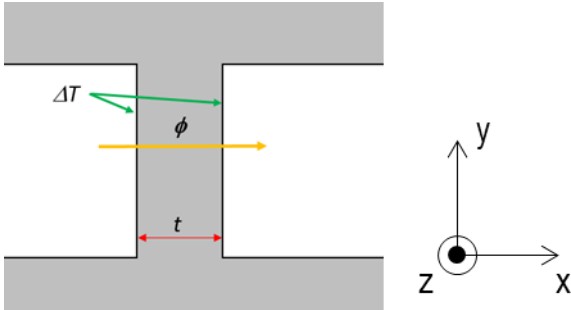

**Figure 4.** Sketch of the quantities used for the evaluation of the effective thermal conductivity.

Since the definition of $R$ reported in Equation (9) is exact only for a 1D problem in the heat flow direction (x in Figure 4), i.e., with infinite walls in <u>both</u> the other two directions (y and z in Figure 4), the value of $k_{\text{eff}}$ is expected to be different from the value of the thermal conductivity of the EUROFER97, as it also accounts for 2D heat diffusion effects. Considering that some of the neighbouring manifold channels contain fluid at the same bulk temperature (e.g. FW1, BZ4 and BZ3, and BZ2 and BZ1 according to the nomenclature in Figure 3), the heat transfer between such manifolds will be negligible, so in this study only two values of $k_{\text{eff}}$ are computed, namely those identified as $k_{\text{eff,FW}}$ and $k_{\text{eff,BZ}}$ in Figure 3.

The mesh size in the 2D computational domain, always obtained using unstructured triangles, has been chosen via a grid independence analysis, whose result is reported in Figure 5. The quantity selected for the grid convergence study is the "effective thermal conductivity" $k_{\text{eff}}$ (see Equation (10) and the discussion above). The value at 0 mesh size is computed with the Richardson extrapolation (RE) method [36]. Three grids are chosen with a refinement ratio $r \cong 1.95$, and the order of convergence of the method is $p \cong 2.06$, computed according to Equation (11) [36]:

$$p = \ln\left(\frac{k_{\text{eff,coarse}} - k_{\text{eff,medium}}}{k_{\text{eff,medium}} - k_{\text{eff,fine}}}\right) / \ln(r), \tag{11}$$

where the subscript coarse, medium and fine refer to the grid refinement; the observed order of convergence is close to the theoretical value of 2 [31]. The grid convergence index (GCI) [36] is reported

in Table 1, showing that the selected mesh (i.e., the finest) is well inside the asymptotic range of convergence; the RE-computed values ($k_{eff,0}$) are $k_{eff,0,BZ}$ = 34.97 W/(m·K) and $k_{eff,0,FW}$ = 33.75 W/(m·K). The selected mesh has about $10^5$ triangular elements, and the maximum diameter of the triangulation is about 3.2 mm; according to the grid convergence study, this mesh guarantees a maximum error of 0.0005% for $k_{eff,BZ}$ and 0.01% for $k_{eff,FW}$. The choice of the triangular elements is justified by the isotropy of the problem at hand, as well as by the presence of thin material layers, both suggesting an unstructured mesh.

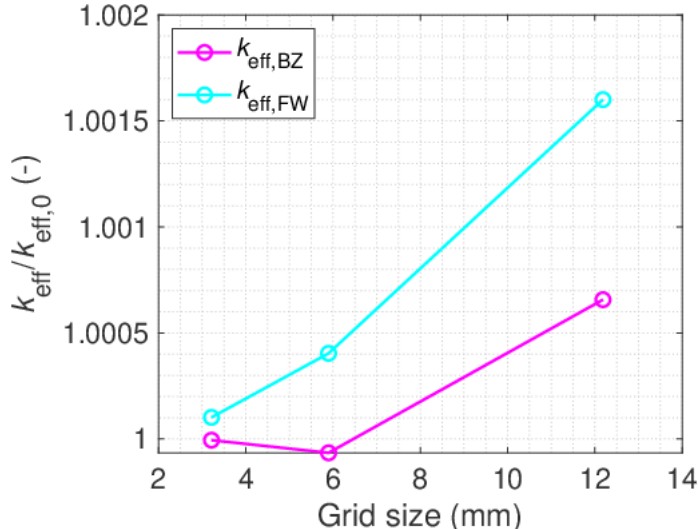

**Figure 5.** Grid convergence study. $k_{eff,0}$ refers to the values at 0 grid size estimated with the Richardson extrapolation (RE) method.

**Table 1.** Results of the grid independence study.[0.]

| Quantity | GCI$_{2,3}$ | GCI$_{1,2}$ | $\lvert r^p \times$GCI$_{1,2}$/GCI$_{2,3}-1\rvert$ |
|---|---|---|---|
| $k_{eff,BZ}$ | 0.008% | 0.0007% | $6 \times 10^{-5}$ |
| $k_{eff,FW}$ | 0.05% | 0.01% | $3 \times 10^{-4}$ |

GCI$_{2,3}$ refers to the medium and coarse grids; GCI$_{1,2}$ refers to the fine and medium grids.

The 1D FV model is implemented with the GETTHEM code, according to the sketch in Figure 6. The code, implemented in the a-causal Modelica language, solves the 1D transient mass, momentum and energy conservation equations, i.e., the 1D inviscid compressible Navier–Stokes equations or Euler Equations (12)–(14) [37]:

$$\frac{\partial \rho}{\partial t} + \frac{\partial \rho v}{\partial x} = 0, \tag{12}$$

$$\frac{\partial \rho v}{\partial t} + \frac{\partial \rho v^2}{\partial x} = -\frac{\partial p}{\partial x}, \tag{13}$$

$$\frac{\partial \rho e}{\partial t} + \frac{\partial \rho e v}{\partial x} = -p\frac{\partial v}{\partial x} + q''', \tag{14}$$

where $t$ is time, $x$ is the direction of the fluid motion, $p$ is the fluid pressure and $e$ is the fluid internal energy. The FV method does not solve the set of equations in the differential form above, but adopts the integral form, obtained integrating Equations (12)–(14) above over the entire domain $\Omega$ and applying the divergence theorem to the terms involving space derivatives. After discretizing the domain in a set of finite control volumes (CV), the FV method enforces the equations above in all the CVs, i.e., solves Equations (15)–(17) [38]:

$$\dot{m}_{in,i} - \dot{m}_{out,i} = \sum_i \frac{dm_i}{dt} = A\sum_i l_i\left(\left.\frac{\partial \rho_i}{\partial p}\right|_h \frac{dp_i}{dt} + \left.\frac{\partial \rho_i}{\partial h}\right|_p \frac{dh_i}{dt}\right), \tag{15}$$

$$\frac{l_i}{A}\frac{d\dot{m}}{dt} + p_{out} - p_{in} = \Delta p_{friction} = \frac{1}{2}f\frac{l_i}{D_h}\frac{\dot{m}_i^2}{\rho_i A^2} \, , \tag{16}$$

$$Al_i\rho_i\frac{dh_i}{dt} + \dot{m}_i(h_{out,i} - h_{in,i}) = q_i''' , \tag{17}$$

$\forall i$, where $\dot{m}$ is the mass flow rate, $i$ is the index of the CV, $m$ is the mass in the CV, $A$ is the area of the cross section of the channel, $l$ is the length of the CV, $h$ is the fluid specific enthalpy and $D_h$ is the hydraulic diameter of the channel.

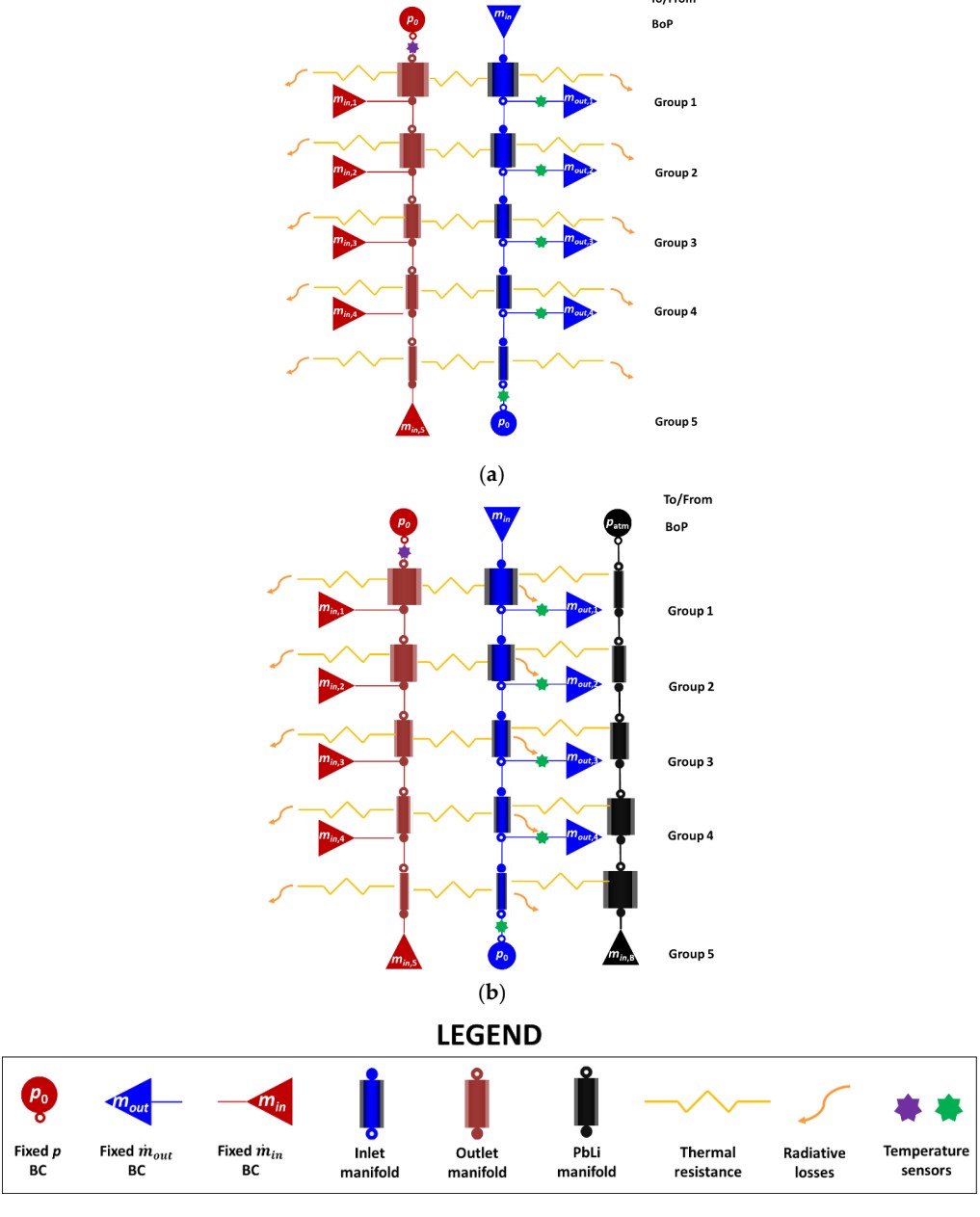

**Figure 6.** Sketch of the 1D GEneral Tokamak THErmal-hydraulic Model (GETTHEM) models: (**a**) first wall (FW) and breeding zone (BZ) manifolds model neglecting heat transfer with PbLi manifolds; (**b**) BZ manifolds model including heat transfer with PbLi manifolds. The thermal resistances before the radiative losses represent the solid material between the manifolds and the vacuum vessel (VV)-facing walls. The different boundary conditions (BCs) refer to different groups of channels, whose naming convention is reported. Different dimensions of the manifold channels are used to represent the variation of the cross section.

According to a strategy already adopted and verified in previous works [39–42], the inlet and outlet manifolds (IM, OM) of the whole BSS are split in 5 portions along the poloidal direction, whose nomenclature is reported in Figure 6. Downstream each portion, part of the mass flow rate is removed from the IM and added to the OM, in order to represent the coolant going to (IM) or coming from (OM) the FW or BZ cooling channels. These are represented as fixed mass flow rate BCs ("$m_{in,1-4}$" and "$m_{out,1-5}$" in Figure 6), as well as the total mass flow rate at the inlet of the IM ("$m_{in}$" in Figure 6); the numerical value of these BCs is reported in Table 2. At the outlet of the IM and OM, fixed pressure BCs are used ("$p_0$" in Figure 6), to set the gauge pressure value to 15.5 MPa. For the PbLi manifolds, a mass flow rate $m_{in,B} \cong 16$ kg/s is imposed at the inlet and atmospheric pressure at the outlet. As both inlet and outlet are routed via the upper port (i.e., from the top region in Figure 2a), the IM and OM are coupled in counter-current. The thermal resistance model depicted in Figure 6 implements the definition in Equation (18):

$$R = \frac{1}{k_{\text{eff}}} \frac{t}{S},$$ (18)

i.e., the same definition as Equation (9) above, but substituting the material thermal conductivity with the effective thermal conductivity; such $k_{\text{eff}}$ values are provided as a correlations, as a function of the wall thickness, according to the results of the 2D FE model.

**Table 2.** Value of the fixed mass flow rate BCs. (Reproduced from [14], Politecnico di Torino: 2018.).

| Quantity | Value (kg/s) |
|---|---|
| $m_{in,1-5}$ and $m_{out,1-4}$ | 5.7 |
| $m_{in}$ | 28.3 |

The model reported in Figure 6a does not include the PbLi manifolds, which are then considered as adiabatic; this approach is useful as it is equivalent to considering the lower bound of the heat transfer to or from the PbLi. Indeed, the knowledge about heat transfer in MHD conditions is limited, in particular if considering complex geometries and coupled channels like those found in the WCLL BSS; however, some efforts are being put in by the WCLL design team to improve understanding of MHD heat transfer and pressure drops [43–45]. To have an idea of the effect of that uncertainty, in the model reported in Figure 6b also the PbLi manifolds are included: this allows for evaluation of the effect of the heat transfer to or from the PbLi. This is done either implementing the correlation (7) reported in [32] or assuming ideal heat transfer, which gives the upper bound and thus, together with the model in Figure 6a, brackets the actual conditions.

## 3. Results

### 3.1. Two-Dimensional (2D) Finite Elements Analysis

As mentioned in Section 2 above, the 2D model is run with the nominal values of the wall thickness with and without radiative boundary conditions on the wall facing the VV; these results are reported in Figure 7. The main feature visible in these results is that the effect of the radiative BC is to cool down the BSS region, and to distort the temperature distribution in the BSS. However, these distortion and temperature reduction effects are rather local to the region close to the VV-facing wall, whereas the temperature isolines and the corresponding values in the region of interest (i.e., in the walls where relevant heat transfer is expected to take place) are nearly identical in the two cases (see insets). This result is confirmed extrapolating the values of $k_{\text{eff,FW}}$ and $k_{\text{eff,BZ}}$ from the two simulations according to Equation (10): as reported in Figure 8, the computed values differ by less than 0.05% (BZ) and 0.03% (FW) for the nominal thickness.

To verify that most of the heat is transferred via the walls of interest (i.e., those highlighted in the insets in Figure 7 below), the power exchanged there is evaluated as a fraction of the total power exchanged by the corresponding manifold channels, for both cases with and without radiative heat

transfer. In all cases, heat exchange through the walls of interest represent >70% of the total power transfer, justifying the implementation of a 1D resistance across such walls, as reported in Figure 9.

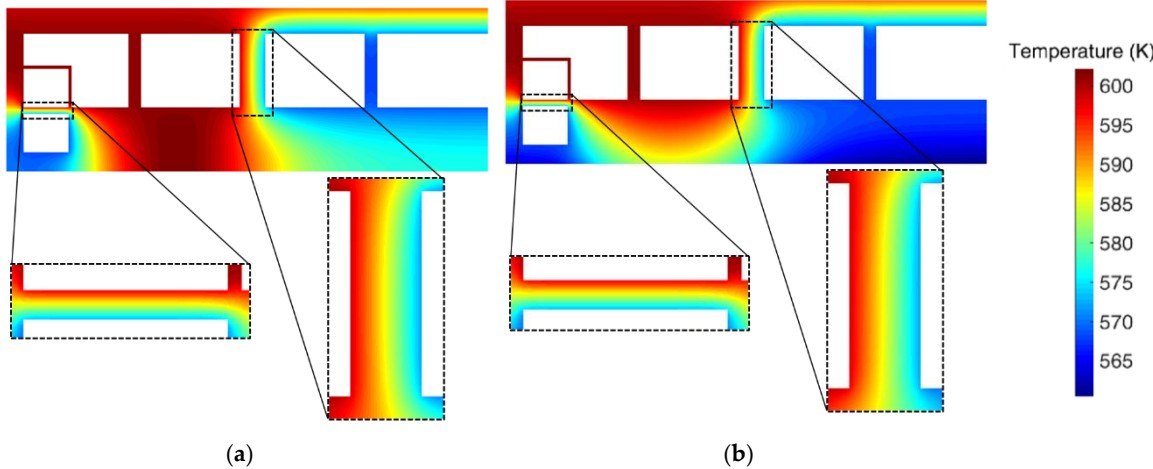

(**a**)                                                                          (**b**)

**Figure 7.** Two-dimensional (2D) temperature map in the manifold region: (**a**) with adiabatic wall; (**b**) with radiative heat transfer. The insets show the detail of the regions of interest.

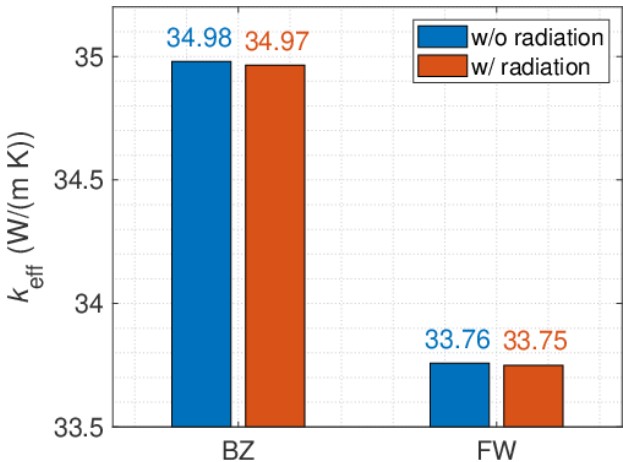

**Figure 8.** Effective thermal conductivities with and without radiative heat transfer on back surface.

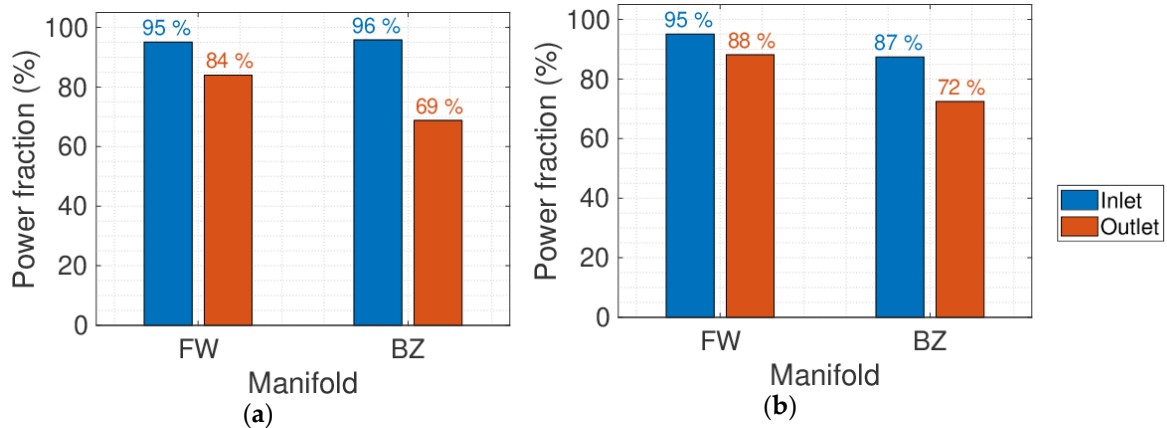

(**a**)                                                                          (**b**)

**Figure 9.** Power transferred through the walls interfacing manifolds at different temperature, as a fraction of the total power exchanged in the corresponding manifold: (**a**) with adiabatic wall; (**b**) with radiative heat transfer.

Then, a set of 2D simulations is performed, varying the thickness of the BSS walls in the range 50–200% of the nominal value. The different values of $k_{eff}$ are reported in Figure 10, where they are also compared to the EUROFER97 thermal conductivity. The deviation from the EUROFER97 value is indeed significant, meaning that 2D effects are not fully negligible, particularly at thicknesses larger than or equal to the nominal value (10 mm for the FW and 40 mm for the BZ, respectively). Moreover, as $t{\to}0$, $k_{eff}{\to}k_{EUROFER97}$, as the wall becomes closer to an infinite slab, thus reducing 2D effects. From Figure 10 it is also clear that this dependency on the thickness of the $k_{eff}$ is linear. Two different correlations are then derived for the effective thermal conductivity, i.e.,

$$k_{eff,BZ} = 32.5 + 0.064t \tag{19}$$

for the BZ ($R^2 = 0.9965$), and

$$k_{eff,FW} = 33.0 + 0.075t \tag{20}$$

for the FW ($R^2 = 0.9997$); in both equations above, $t$ is the thickness in millimeters, and the unit of $k_{eff}$ is W/(m·K). To verify the independence of the computed $k_{eff}$ values on the fluid HTC, as mentioned in Section 2 above, the model is run varying the HTC in the range ±20%, estimating the corresponding variation of the $k_{eff}$; such variation is always <6% (see the filled regions in Figure 10).

The correlations reported in Equations (19) and (20) above are then used to evaluate different values of the global thermal resistance between two manifold channels, according to Equation (18), including the value of the heat-transfer surface. This thermal resistance can then be implemented in the GETTHEM models sketched in Figure 6 above. Note that, in view of the linear dependency of $k_{eff}$ on $t$, the thermal resistance will saturate for increasing values of the wall thickness.

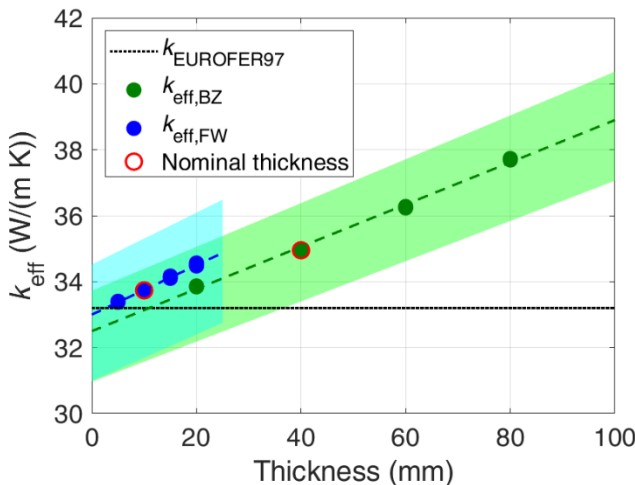

**Figure 10.** Computed values of the effective thermal conductivity as a function of the thickness; the fitting curves are also reported. The filled regions represent the relative variation of $k_{eff}$ when the convective heat transfer coefficient (HTC) varies by ±20%.

### 3.2. One-Dimensional (1D) Finite Volume Analysis

Starting from the 2D results above, the GETTHEM model is run parametrically, to assess the effect of a variation of the wall thickness from the (current) design value. The thickness of the walls all along the manifolds is then varied in the same range of the 2D analyses (50% to 200% of the nominal value). As output, the water temperature at the inlet of the FW and BZ cooling channels (i.e., the green stars in Figure 6), and at the outlet of the outlet manifold (i.e., the purple stars in Figure 6) is evaluated. As mentioned in the Introduction, the first of these two variables is important to guarantee adequate cooling of the components (which are designed assuming a coolant inlet temperature of 295 °C), whereas the latter (whose design value is 328 °C) is important as it affects the plant efficiency. Thanks

to the "hybrid" approach presented above, the model runs very fast (7 ms for a single simulation on an Intel®Core™ i7-8850H CPU @2.60 GHz, single core), thus parametric analyses like that presented here are feasible. For the same reason, this analysis is carried out in three steps:

1.  Without both radiative heat transfer towards VV and heat transfer towards the PbLi manifolds;
2.  With radiative heat transfer towards VV and without heat transfer towards the PbLi manifolds;
3.  With both radiative heat transfer towards VV and heat transfer towards the PbLi manifolds.

The steps above allow us to gain an idea of the impact of the assumptions discussed in Section 2 above concerning the radiative heat transfer to the VV, as well as the impact of the uncertainty about the convective heat transfer towards the PbLi.

In addition, to verify the correctness of the implementation of the correlations derived from the 2D studies in the 1D model, the heat flux computed by the two models at the equatorial outboard midplane (i.e., on the plane where the 2D model has been developed) is compared in Figure 11 for the nominal thickness. The relative discrepancy of the two results is about 5% for the BZ and about 3% for the FW, with the 1D model underestimating the heat flux. This discrepancy is justified considering the accuracy of the GETTHEM model (estimated to be about 3% [46]), and that the fluids at the equatorial midplane already underwent some heat transfer, so the temperatures of the two fluids in the 1D model will be different from those imposed in the 2D model. In particular, the temperature difference between the manifolds is reduced as the effect of this heat transfer, thus reducing the heat flux, which is causing the underestimation in the 1D result.

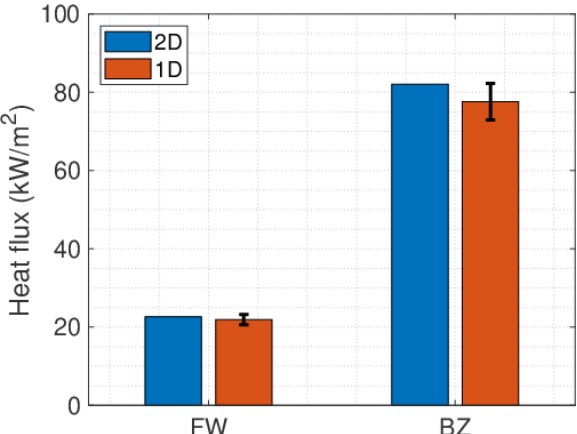

**Figure 11.** Heat flux across the FW and BZ resistances as computed by the 1D and 2D models. (The error on the 2D results is negligible and thus not shown.).

The relative importance of the conductive heat transfer with respect to the convective heat transfer is estimated by computing the total thermal resistances per unit surface as the series of the three resistances (convective in the IM, conductive in the wall, convective in the OM). Recall that the convective thermal resistance per unit surface is defined according to Equation (21),

$$R''_{conv} = 1/h, \tag{21}$$

whereas the conductive thermal resistance per unit surface, starting from Equation (9), is defined according to Equation (22):

$$R''_{cond} = t/k; \tag{22}$$

as the three resistances are in series, the total thermal resistance $R''_{tot}$ is simply the sum of the three. An overall HTC (or "thermal transmittance") $U$ can then be computed as $1/R''_{tot}$. This analysis is reported in Figure 12, where it is clear how the transmittance in the BZ case is much lower with respect to the FW case. This is due to both a larger thickness of the wall and a lower coolant velocity;

consequently, the heat transfer between the BZ manifolds is expected to be smaller with respect to that between the FW manifolds. The most important contribution to the total thermal resistance is in both cases due to the conductive resistance, which represents between 58% and 78% of the total resistance in the FW case and between 64% and 81% in the BZ case.

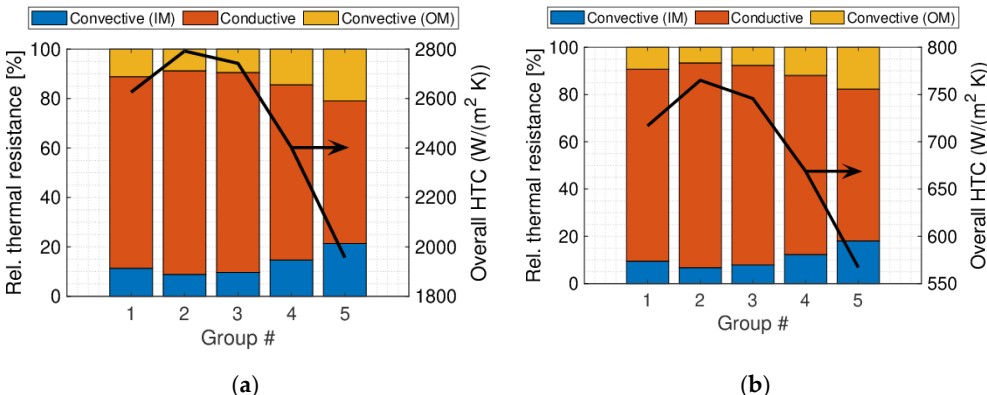

(a)  (b)

**Figure 12.** Distribution in the poloidal direction of the overall HTC (right axis) and of the relative contribution of the three terms to the total thermal resistance (left axis): (**a**) FW manifolds; (**b**) BZ manifolds.

### 3.2.1. No Radiation, Adiabatic PbLi Manifolds

The results for the first scenario, i.e., without both radiation and heat transfer to the PbLi, are reported in Figure 13, which shows the difference between the computed channel inlet temperature and the nominal value (i.e., $T_{in,computed} - T_{in,nominal} = T_{in,computed} - 295\,°C$), and in Figure 14, which reports the difference between the nominal outlet temperature and the computed value (i.e., $T_{out,nominal} - T_{out,computed} = 328\,°C - T_{out,computed}$). In this first scenario the reduction of the coolant inlet temperature is below 1.5 °C for the FW and 0.5 °C for the BZ, so it is not expected to affect significantly the cooling performance. Even smaller deviations are found when looking at the coolant outlet temperature from the tokamak, which is below 1 °C for the FW and even below 0.5 °C for the BZ (which is then negligible according to the accuracy of the model).

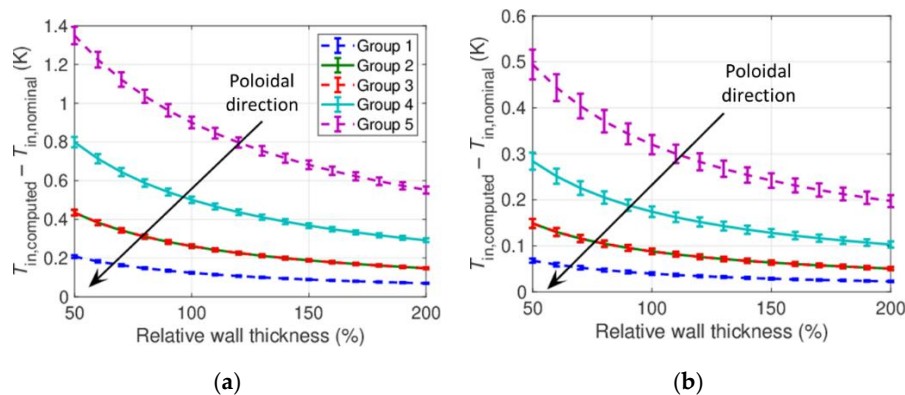

(a)  (b)

**Figure 13.** Distribution of the difference between the computed channel inlet temperature and its design value (295 °C), as a function of the wall thickness, for the case with no radiation and adiabatic PbLi manifolds: (**a**) FW manifolds; (**b**) BZ manifolds. Different lines represent different groups of channels in the poloidal direction, according to the lumping strategy (see Section 2 above and the nomenclature in Figure 6).

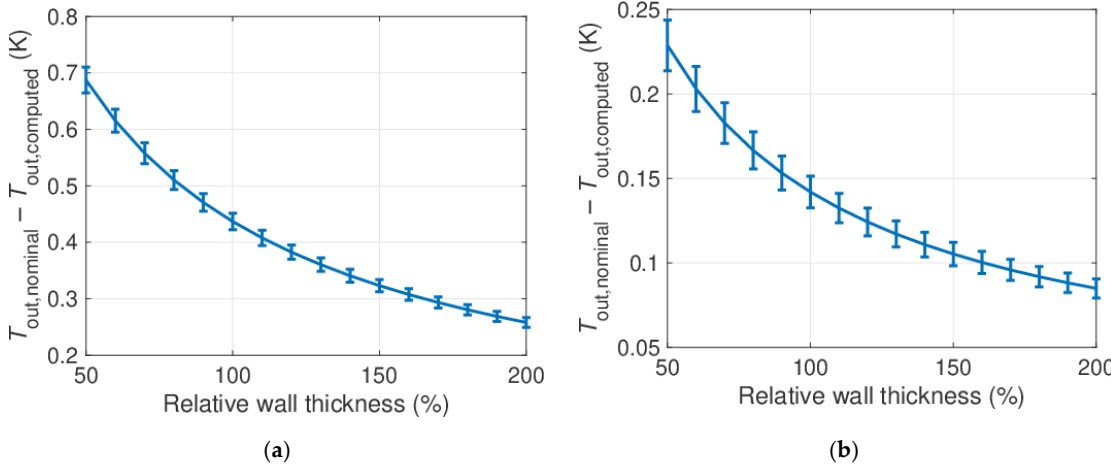

**Figure 14.** Difference between design outlet temperature (328 °C) and its computed value, as a function of the wall thickness, for the case with no radiation and adiabatic PbLi manifolds: (**a**) FW manifolds; (**b**) BZ manifolds.

### 3.2.2. Radiation, Adiabatic PbLi Manifolds

The effect of the inclusion of the VV radiative losses is to cool down the solid BSS region, and thus to cool down also the manifolds. Thus, the inlet water temperature increase will be mitigated, whereas the outlet water temperature reduction will be worsened. Similar to the previous Section, the results are reported in terms of $T_{in,computed} - 295\,°C$ (Figure 15) and $328\,°C - T_{out,computed}$ (Figure 16).

In the case of the FW circuit, the temperature reduction at the inlet of the FW channels (Figure 15a) due to radiative heat transfer to the VV is compensating almost completely the heat transfer with the outlet manifold. The estimated temperature increase is in fact near zero or even negative (i.e., temperature reduction). For the nominal thickness, indeed, all channel groups show a deviation from the nominal value within 0.1 °C, and in any case it is below 0.5 °C. For the BZ circuit (Figure 15b) the reduction of such inlet temperature deviation is smaller in absolute terms, as the BZ inlet manifolds are located further from the VV facing wall with respect to the FW one (see Figures 2d and 3). However, as the temperature deviation was already small in the case without radiation, the radiation towards VV is sufficient to have an inlet temperature always (marginally) below the nominal value.

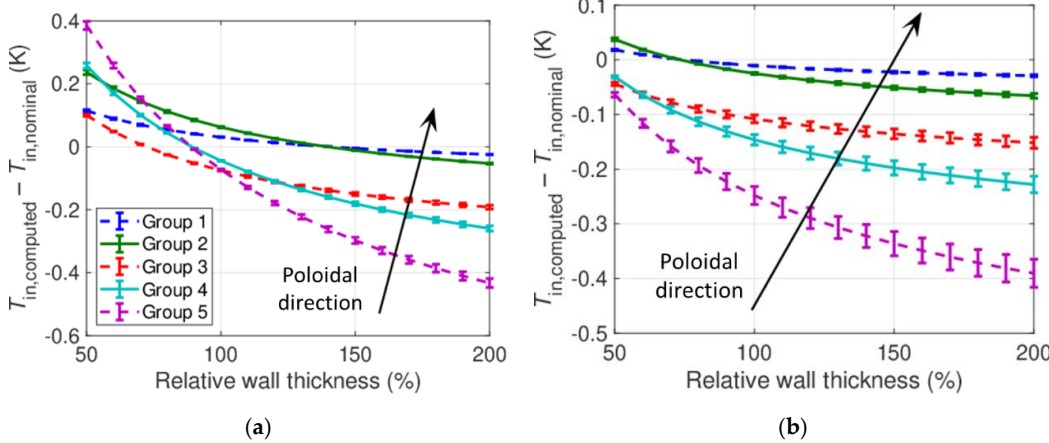

**Figure 15.** Distribution of the difference between the computed channel inlet temperature and its design value (295 °C), as a function of the wall thickness, for the case with radiation and adiabatic PbLi manifolds: (**a**) FW manifolds; (**b**) BZ manifolds. Different lines represent different groups of channels in the poloidal direction, according to the lumping strategy (see Section 2 above and the nomenclature in Figure 6). Negative numbers mean that the computed inlet temperature is smaller than its nominal value.

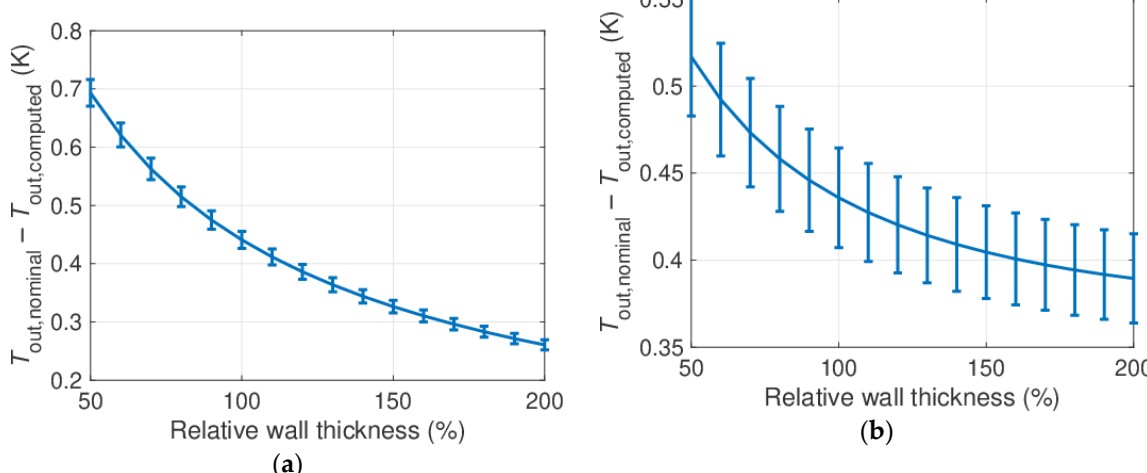

**Figure 16.** Difference between design outlet temperature (328 °C) and its computed value, as a function of the wall thickness, for the case with radiation and adiabatic PbLi manifolds: (**a**) FW manifolds; (**b**) BZ manifolds.

Looking at the outlet temperature reduction, in the FW case (Figure 16a) the result is nearly identical with respect to that obtained when neglecting radiation. This is justified by the geometry, see Figures 2d and 3, as the FW outlet manifold is totally "shielded" from the VV facing wall by the FW inlet manifold. As the FW inlet manifold temperature is reduced, an effect could be seen also on the outlet manifold due to enhanced heat transfer; however, since the FW inlet temperature reduction is small (see Figure 15a), this effect is negligible. A larger effect is instead visible in the BZ outlet manifold (Figure 16b), as no "shielding" manifold is found between it and the VV facing wall; indeed, the temperature deviation from the "no radiation" scenario is more than doubled; however, it is still always <~0.5 °C, so no visible effect is expected on the plant efficiency.

### 3.2.3. Radiation, Heat Transfer with PbLi

To include the effect of the heat transfer with the PbLi manifolds, the GETTHEM model in use is that sketched in Figure 6b. Such an effect is analyzed only for the BZ inlet manifold, as the water in the outlet manifolds (both FW and BZ) will be at a temperature very close to the PbLi temperature (see Section 2 above), and so heat transfer will be negligible therein. In addition, the FW inlet manifold will be shielded by the outlet manifolds, see Figures 2d and 3, similarly to the shielding effect discussed for the FW outlet manifold in Section 3.2.2 above. This effect is evaluated for the nominal thickness value and reported in Figure 17a, comparing the case with radiative losses only (Section 3.2.2 above) against the case including both radiative losses and the PbLi manifolds.

The presence of the PbLi manifold at a temperature larger than the water inlet temperature is expected to increase the inlet temperature; nevertheless, the relatively low PbLi velocity causes the HTC on the PbLi side to be very low (in the range of about 1–10 W/(m$^2$·K)); indeed the difference with respect to the case with adiabatic PbLi manifolds is well below model accuracy (see Figure 17a).

In view of the large uncertainty about the MHD heat transfer, as mentioned in Section 2 above, the simulation is re-run assuming ideal heat transfer (i.e., infinite HTC), such that the temperature of the solid surface on the PbLi side is assumed to be equal to the PbLi bulk temperature. This condition is the upper bound of the heat transfer, and if compared to the case with adiabatic PbLi manifolds (which is the lower bound) allows evaluating minimum and maximum temperature deviation at channel inlet. Note however that the actual condition is expected to be closer to the adiabatic case, in view of the low expected HTC on PbLi side.

The result of this extremely conservative scenario is reported in Figure 17b: the total temperature variation due to the presence of the PbLi manifolds is, in the worst case, about 0.2 °C, so that the effect

is to compensate the temperature reduction due to radiative losses. Indeed, in all the channel groups the inlet temperature is equal to the nominal value up to deviations <0.1 °C, which are below model accuracy. More in detail, in the first two groups of channels the presence of the PbLi manifolds reverts the beneficial effect of the radiative heat transfer, as the computed inlet temperature is (marginally) above the nominal value. In the remaining three groups, instead, the effect of the radiative losses to the VV is still predominant and the computed inlet temperature is below the nominal value (which is a beneficial effect).

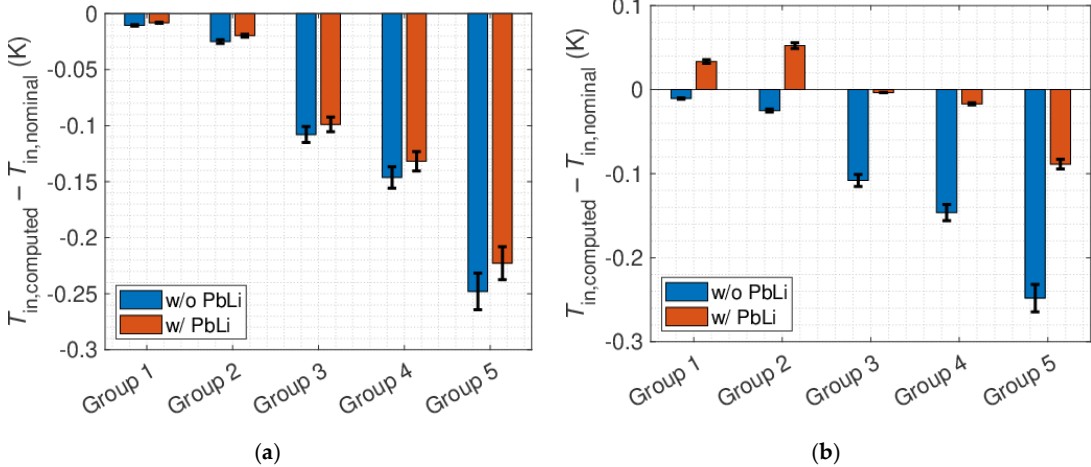

(**a**)  (**b**)

**Figure 17.** Distribution of the difference between the computed channel inlet temperature and its design value (295 °C), for the case without and with heat transfer with the PbLi manifolds, for the BZ circuit: (**a**) HTC from correlation; (**b**) infinite HTC. Negative numbers mean that the computed inlet temperature is smaller than its nominal value. (Refer to Figure 6 for the channel group nomenclature.).

## 4. Conclusions

A 1D model has been set up for the thermal-hydraulic analysis of the water manifolds of the Water-Cooled Lithium-Lead breeding blanket for the EU DEMO. Such a model includes 2D heat-transfer effects in the back supporting structure by implementing correlations derived from a set of 2D analyses; this approach allows computing the total heat transfer across manifolds at different temperatures in a 3D object, but with a dramatic reduction of the computational cost needed.

The model has been applied to estimate the heat transfer occurring between water inlet and outlet manifolds, which is to be minimized in order to avoid (1) a reduction of the cooling performance and (2) a reduction of the plant efficiency. Thanks to the reduced computational cost, a parametric scan has been performed, varying the thickness of the walls separating the different manifolds, and including different effects such as radiation heat transfer with the VV (which is at a much lower temperature) and heat exchange with the PbLi manifolds.

The results highlighted that even in the worst case considered in the present work, such heat transfer is negligible, as it causes a temperature deviation from nominal operating conditions below 1.5 °C, i.e., less than 5% of the design temperature increase in the machine; moreover, in some of the analyzed scenarios, the combined effect is even beneficial (although still below 1% of the design temperature variation).

The proposed approach of combining 2D analyses to determine correlations for a 1D model might be applied in similar contexts. It allows development of fast-running 1D models which partially include also 2D effects, provided the parameters stay within the range of validity of the developed correlations, reducing the need for computationally expensive 3D analyses. Further developments in the methodology could include the most general case where also axial heat conduction should be considered (which is negligible here), i.e., the development of heat conduction correlations starting from a 3D model rather than a 2D model. In addition, the validity of the approach, developed for a steady-state case, in a transient scenario, shall be assessed.

**Author Contributions:** Conceptualization, A.D.N.; methodology, A.B., A.F., L.S.; software, A.B., A.F.; writing—original draft preparation, A.F.; writing—review and editing, A.B., A.D.N., A.F., L.S.; visualization, A.F.; supervision, L.S.; funding acquisition, A.D.N. All authors have read and agreed to the published version of the manuscript.

**Funding:** This research was funded by the Euratom research and training programme 2014–2018 and 2019–2020, grant number 633053. The APC was funded by the Euratom research and training programme 2014–2018 and 2019–2020, grant number 633053.

**Acknowledgments:** This work has been carried out within the framework of the EUROfusion Consortium. The views and opinions expressed herein do not necessarily reflect those of the European Commission.

**Conflicts of Interest:** The authors declare no conflict of interest. The funders had no role in the design of the study; in the collection, analyses, or interpretation of data; in the writing of the manuscript; or in the decision to publish the results.

## Nomenclature

**Abbreviations.**

| | |
|---|---|
| BB | Breeding blanket |
| BC | Boundary condition |
| BoP | Balance-of-plant |
| BSS | Back supporting structure |
| BZ | Breeding zone |
| CFD | Computational fluid-dynamics |
| CV | Control volume |
| DWT | Double-walled tubes |
| ENEA | Italian national agency for new technologies, energy and sustainable economic development |
| EU DEMO | European demonstration fusion power reactor |
| FE | Finite element |
| FV | Finite volume |
| FW | First wall |
| GCI | Grid convergence index |
| GETTHEM | GEneral Tokamak THErmal-hydraulic Model |
| HTC | Heat transfer coefficient |
| IAPWS | International association for the properties of water and steam |
| IF97 | Industrial formulation '97 |
| IM | Inlet manifold |
| MHD | Magnetohydrodynamics |
| OM | Outlet manifold |
| RE | Richardson extrapolation |
| TBM | Test blanket module |
| VV | Vacuum vessel |
| WCLL | Water-Cooled Lithium-Lead |

**Symbols.**

| | | |
|---|---|---|
| $A$ | Area of cross section | $m^2$ |
| $c_p$ | Specific heat at constant pressure | $J/(kg \cdot K)$ |
| $D$ | Diameter | m |
| $e$ | Specific internal energy | J/kg |
| $f$ | Darcy friction factor | - |
| $f_v$ | View factor | - |
| $h$ | Heat transfer coefficient, specific enthalpy | $W/(m^2 \cdot K)$, J/kg |
| $i$ | Control volume index | - |
| $k$ | Thermal conductivity | $W/(m\ K)$ |
| $L$ | Characteristic length | m |
| $m$ | Mass | kg |
| $\dot{m}$ | Mass flow rate | kg/s |

**Symbols.**

| | | |
|---|---|---|
| $Nu$ | Nusselt number | - |
| $p$ | Order of convergence, pressure | -, Pa |
| $Pe$ | Péclet number | - |
| $Pr$ | Prandtl number | - |
| $Q$ | Heat flow | W |
| $q''$ | Heat flux | W/m$^2$ |
| $q'''$ | Volumetric power | W/m$^3$ |
| $R$ | Thermal resistance | K/W |
| $r$ | Refinement ratio | - |
| $R''$ | Thermal resistance per unit surface | K·m$^2$/W |
| $R^2$ | Coefficient of determination | - |
| $Re$ | Reynolds number | - |
| $S$ | Surface area | m$^2$ |
| $T$ | Temperature | K |
| $\Delta T$ | Temperature difference | K |
| $T$ | Thickness, time | m, s |
| $U$ | Thermal transmittance (overall heat transfer coefficient) | W/(m$^2$·K) |
| $v$ | Velocity | m/s |
| $w$ | Test function | - |
| $x$ | Direction along fluid motion | m |

**Greek.**

| | | |
|---|---|---|
| $\varepsilon$ | Emissivity | - |
| $\mu$ | Dynamic viscosity | Pa s |
| $\rho$ | Density | kg/m$^3$ |
| $\sigma$ | Stefan–Boltzmann constant | W/(m$^2$·K$^4$) |
| $\phi$ | Heat flux | W/m$^2$ |
| $\Omega$ | Domain | - |

**Subscripts.**

| | |
|---|---|
| 0 | Imposed (Dirichlet BC), RE-computed value |
| 1 | Fine grid |
| 2 | Medium grid |
| 3 | Coarse grid |
| BZ | Breeding zone |
| coarse | Coarse grid |
| computed | Value computed by the code |
| D | Dirichlet |
| eff | Effective |
| fine | Fine grid |
| FW | First wall |
| g | Fluid |
| h | Hydraulic |
| i | Control volume index |
| in | Inlet |
| medium | Medium grid |
| N | Neumann |
| nominal | Nominal or design value |
| out | Outlet |
| R,c | Convective Robin |
| R,r | Radiative Robin |
| s | Slug |
| VV | Vacuum vessel |

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
