# Peer review of "Hybrid 1D + 2D Modelling for the Assessment of the Heat Transfer in the EU DEMO Water-Cooled Lithium-Lead Manifolds"

_energies, doi:10.3390/en13143525_

Round 1

Reviewer 1 Report

The topic of the paper is interesting and within the scope of the journal Energies. The originality of the work is acceptable. Most of the Figures are legible and unambiguous. I recommend only minor revisions before the publication of the manuscript.

  • Abstract – Please start the Abstract with the general sentence that will describe the background of the research.
  • Abstract – Please also underline the significance and originality of the research in this section. Why the presented results are useful and who can use them?
  • Keywords – Don’t use abbreviations that readers would not quickly recognize. Change “WCLL” to “Water-Cooled Lithium-Lead (WCLL)”.
  • Introduction - I have doubts about the order of literature citations in the main text. For example, the first number that appears in the main text is [2]. Number [1] is only on the next page. In my opinion, number [1] should be added when referring to Figure 1 or the order of references should be changed. Similarly, in the case of reference to item [7] in the title of Figure 2. The number [7] should be changed to [6] and added in line 37.
  • Introduction – The novelty of the work has been described in just one sentence. In my opinion, the significance and originality of the study should be more emphasized.
  • Materials and Methods – Please add references to the literature for the equations (e.g. eq. (9), (10), (11)-(17)).
  • Results – Although I have no objections to most Figures, Figure 8 is an exception. I suggest adding specific values in this Figure, because the differences between the effective thermal conductivity values are relatively small. Therefore, it is difficult to determine them based on the height of the columns. Specific values can also be added in Figure 9.
  • Conclusions – In the Conclusions section, the directions for further research should be defined in detail. It is also necessary to specify the limitations that affect the results.
  • References – Please adapt the formatting of the references to the mdpi guidelines. For example, the article citations lack commas (after the year and volume number).
  • Due to the fact that there are a lot of symbols in the article, I suggest adding Nomenclature sections in the manuscript.

Best regards,

Reviewer 2 Report

1-  Check the manuscript and remove typos from the text.

2-  Fig. 1: Can authors elaborate more on the items and units and their contributions? It is not clear how the system works.

3-  Equations 2-5 need proper support from literature, or derivation should be added to the paper.

4-  Can authors add error bars to Fig 5? How accurate the keff/keff,0 is? Very critical parameter.

5-  Add more quantitative results to the conclusion and also the abstract part.

Reviewer 3 Report

This manuscript title, 1D + 2D hybrid modeling for the assessment of the heat transfer in the EU DEMO Water-Cooled Lithium- Lead manifolds, presents a reduced-order computational model to calculate heat transfer characteristics between the hot and cold manifolds of EU DEMO BSS. The study seems interesting, however, the methodology and novelty of the work in not clear. Very long sentences have bee used, the content of the paper is unclear and hard to understand. The following concerns on the current work must be addressed before considering for a publication.  

  1. The introduction section should not only include previous work. The author should provide a constant comparison with previous work to highlight how the present study is novel. The last paragraph of the introduction section does not provide a clear justification for the novelty of the work and requires a revision.
  2. Line 194, why L is taken as half of the channel length and not equal to the hydraulic diameter of the channel?
  3. . Please review equations 12-14. Are these equations for an inviscid and incompressible flow ??
  4. Did the authors compare the computational cost of 3D RANS simulations with the developed reduced-order model? Please report how much computation cost has been reduced and at what cost i.e., loss in accuracy of the solution.
  5. Revise the paper thoroughly to improve the language and grammar of the paper.

Reviewer 4 Report

The blanket is one of the important components within a fusion power plant. Froio et al. set up models for the thermal-hydraulic analysis of the water manifolds of the Water-Cooled Lithium-Lead Breeding Blanket for the EU DEMO, allowing estimation of the total heat transfer across manifolds. The authors further proposed an approach of combining 2D analyses to determine correlations for a 1D model which allows the development of fast-running 1D models. Overall, the manuscript is well-organized. The study would improve the understanding of heat transfer of the WCLL breeding blanket, providing insights into the design of the Breeding Blanket for the EU DEMO. The manuscript would be of interest to a general audience. I have a few suggestions for the authors:

  1. Figure 5 shows the effective thermal conductivity for different grid sizes. In the case of FW, why does keff/keff,0 first decrease then increase as grid size increases?
  2. Figure 17 shows the distribution of the difference between the computed channel inlet temperature and its design value (295 °C) for the BZ circuit. In b), why do the temperature differences of groups 1 and 2 with PbLi show values larger than zero, while the differences of the other groups show values smaller than zero?

Round 2

Reviewer 2 Report

Nomenclature parts do not have any units

Author Response

Units have been added.

Reviewer 3 Report

The authors have made a good effort and improved the manuscript significantly. I recommend the paper for publication in its present form. 

Author Response

We thank the reviewers for their precious comments.